# Preliminary Study on the Thermal Behavior and Chemical-Physical Characteristics of Woody Biomass as Solid Biofuels

Max J. A. Romero, Daniele Duca, Vittorio Maceratesi, Sara Di Stefano, Carmine De Francesco and Giuseppe Toscano *

Department of Agricultural, Food and Environmental Sciences, Università Politecnica Delle Marche, Via Brecce Bianche, 60131 Ancona, Italy
* Correspondence: g.toscano@univpm.it

**Abstract:** The chemical composition of woody biomass directly influences its thermal degradation and, subsequently, the selection of processes and technologies used for its conversion into energy or value-added products. Thus, the present study aimed to evaluate the thermal behavior and chemical-physical characteristics of three different woody biomass species (hardwood, softwood and chemically-treated wood) using thermogravimetric and characterization analysis based on ISO 16948, ISO 18125 and ISO 18122 methods. The main findings show that the most significant trend of mass loss, around 70%, in the thermal degradation of the different species of woody biomass occurred between 150 °C and 500 °C and that the residual mass at 650 °C was between 13% and 24%. Although the three species of woody biomass showed a high average energy content (19.60 MJ/kg), softwood samples had a more stable thermal degradation than hardwoods and chemically-treated woods.

**Keywords:** woody thermal degradation; thermogravimetric analysis; solid biofuel analysis; thermochemical process; combustion

## 1. Introduction

According to the latest data, the electricity sector accounted for 36% of all energy-related emissions in 2020. Most electricity produced worldwide comes from coal, which is responsible for around three-quarters of the sector's $CO_2$ emissions. To this end, several initiatives have been deployed to increase the share of renewable energies to 40–70% by 2050 and achieve a rapid transition to decarbonization in the electricity sector [1]. One of these initiatives is the production of biofuels, which have the potential to provide at least 25% of the world's projected energy needs by 2035 [2]. Particularly, lignocellulosic biomass represents an abundant and carbon-neutral non-food resource that is being intensively investigated as a raw material for producing novel solid biofuels [3]. According to ISO 17225, the main origin-based solid biofuel groups are woody biomass, herbaceous biomass, fruit biomass, aquatic biomass and blends and mixtures. Woody biomass includes forest, plantation and other virgin wood and by-products or residues from the wood processing industry. Wood by-products and wood residues from industrial production can be chemically untreated. For example, residues from debarking, sawing, size reduction, shaping and pressing. On the other hand, chemically-treated woody biomass includes wood residues from wood processing and the production of panels and furniture (glued, painted, coated, lacquered or otherwise treated wood), provided they do not contain heavy metals or halogenated organic compounds from treatment with wood preservatives or coating [4]. However, in several European countries, including Italy, treated wood residues such as glue-laminated wood, oriented strand board (OSB), plywood and chipboard are normatively considered waste, and they do not allow their utilization for energy application because they could contain unwanted or harmful compounds [5]. In fact, according to the

Italian Legislative Decree 156/2006, a waste is defined as "any substance or object that an owner discards or has an intention or obligation to discard" [6].

Woody biomass energy can be extracted and made available directly or indirectly with low-emission generation technologies such as combined heat and power (CHP) systems. Although cogeneration accounted for 11% of the EU's electricity and 15% of its heat in 2020, it is estimated that it will grow to 20% in 2030, when it will avoid 350 Mt $CO_2$ of emissions and account for 25% of the EU's heat [7]. In addition, the EU's strategic long-term vision for a climate-neutral economy by 2050 stresses the importance of an integrated energy system approach to achieve deep emissions reductions [8,9]. The CHP systems are mainly composed of a combustion device that transforms the chemical energy stored within the fuel into heat and a generator that subsequently transforms part of the heat into electricity. Regarding the combustion process, CHP systems can be mainly classified in technologies based on an external combustion process such as steam turbines, Stirling engines and ORC systems and technologies based on an internal combustion process such as internal combustion engines (ICEs) and gas turbines [10,11]. Since ICEs cannot directly use solid biomass as fuel, some CHP systems may also include a third step in which solid biomass is converted into liquid or gaseous biofuels through additional thermochemical (e.g., pyrolysis, gasification and liquefaction) or biochemical processes (e.g., anaerobic digestion and fermentation) [12]. Although large-scale CHP systems are preferred due to a better net profit with respect to the investment, it was reported that using micro- and small-scale CHP systems for valorizing biomass resources in local areas could also be economically feasible [13–16]. The European Directive 2004/8 EC defines micro-scale CHP systems and small-scale CHP systems as systems having electrical capacities of less than 50 kW and ranging from 50 kW to 1 MW, respectively. At present, it is typical to find micro- and small-scale combined heat and power (CHP) systems on the market based on the direct combustion of liquid or gaseous fuels such as refined vegetable oils (RVO) and bio-oils [17–24], biogas [25–29] and syngas [30–36]. On the other hand, more recent studies propose making biomass energy available using promising technologies such as fuel cell technologies. For example, the direct carbon fuel cell (DCFC) is a high-temperature (usually >500 °C) fuel cell technology that can convert solid biocarbon fuel directly into electricity without an extra reforming process, with an efficiency greater than 80%, unlike other fuel cells. Hence, compared to a current coal-burning power plant, the direct carbon fuel cell has a lower $CO_2$ emission at the same amount of electricity generation [37]. Moreover, Photo Fuel Cell (PFC) systems have been reported for photocatalytic electricity generation from lignocellulosic biomass. The peak power density (Pmax) of PFC systems reported is in the range of 0.01–5 mW/$cm^2$, being higher than those obtained by microbial fuel cells (MFCs) but significantly lower than those obtained by proton exchange membrane fuel cells and DCFCs. Efficient lignocellulose conversion based on photocatalysis is a promising topic because it uses sustainable solar energy and mild process conditions. However, it is crucial to develop photoelectrodes that improve their performance [38]. Additionally, the use of woody biomass can be extended to other industrial sectors such as food, cosmetics, pharmaceuticals, electronics and the environment through the production of value-added derivatives. For example, the lignin-first approach has recently emerged as a new biorefinery model that proposes using the entire lignocellulosic biomass to disassemble lignin prior to cellulose and hemicellulose valorization [39–41].

In this context, to enable the development of technologies that more efficiently convert woody biomass into energy, biofuels and value-added chemicals, it is important to increase the knowledge of its chemical-physical characteristics. Woody biomass mainly comprises three fractions, namely cellulose, hemicellulose, and lignin, which approximately represent 40–50%, 25–35% and 15–20% of its weight, respectively, depending on the source (hardwood, softwood, or grasses) [42]. Cellulose and hemicellulose are polymers composed of glucose monomers linked mainly through -1,4-glycosidic bonds [43]. However, cellulose has a rigid structure due to the main polymer chain's cyclic structure and strong intermolecular hydrogen bonds between the hydroxyl groups, while hemicellulose has

an amorphous as well as a random structure organized in branched chains [44]. Both compounds play an important role in woody biomass's stability, resistance and insolubility to most organic solvents and water. Moreover, lignin is an aromatic polymer that acts similar to glue by filling the gap between the cellulose and hemicellulose and is composed of primary lignin monomers such as coniferyl alcohol, sinapyl alcohol and coumaryl alcohol (C9–C11 monolignols) [45]. Due to their different chemical structures, hemicellulose and cellulose have different degradation temperatures in the ranges of 200–300 °C and 300–400 °C, respectively. However, both compounds pyrolyze through similar routes: at low temperatures, dehydration is favored, forming water, gases, char, and further acids and anhydrides; at higher temperatures, depolymerization forms levoglucosan (dehydrated analogue of glucose) and other anhydrosugars, alongside levoglucosene (the dehydration product of levoglucosan), furans and other volatile organics. Lignin follows the same pyrolysis pattern but has greater thermal stability, and its degradation temperature ranges between 200 and 500 °C [37]. Therefore, the chemical composition of woody biomass directly influences its thermal degradation and, subsequently, the selection of processes and technologies used for its conversion into energy or value-added products [46–48].

The European standards for solid biofuels (EN ISO 17225-series) define different quality classes of solid biofuels based on qualitative attributes (origin and source of the material) and chemical-physical parameters. The aim of this work is to investigate the thermal behavior and chemical-physical characteristics of the most common European woody biomass species used in the energy pellet sector by using thermogravimetric analysis and characterization analysis (heating value, elemental composition (CHNO and ash content).

Ding et al. [49] focused on the thermal decomposition characteristics and kinetic mechanisms of hardwood and softwood by thermogravimetric analysis in nitrogen. The obtained activation energy of softwood was greater than that of hardwood during the whole pyrolysis process, and the kinetic mechanisms for both hardwood and softwood can be summed up as a diffusion mechanism followed by a reaction order mechanism. Yao et al. [50] calculated the apparent activation energy of hardwood and softwood in nitrogen atmosphere, and the mean values were approximately 155 and 160 kJ/mol for hardwood and softwood, respectively [51]. It should be noted that the thermal degradation characteristics, activation energy, and kinetic mechanisms of woody biomass under oxidative atmosphere are quite diverse from those under inert atmosphere. In fact, the thermal degradation of solid materials in oxidizing atmosphere is more complicated because the presence of oxidants (air, oxygen, etc.) will produce heterogeneous reactions between oxygen and solid reactants and homogeneous reactions between oxygen and volatiles. However, softwood decomposition began and ended at lower temperatures than hardwood in air atmosphere. In oxidizing atmosphere two diverse peaks and one shoulder appeared on the reaction rate curves for both hardwood and softwood. The maximal reaction rate of hardwood was larger than that of softwood [52–54]

Thermogravimetric analysis (TGA) is a reliable, low-cost and fast method for investigating the thermal decomposition of biomass in oxidizing (combustion and gasification) and inert atmospheres (pyrolysis and torrefaction) [49–54]. TGA is a method that consists of submitting a sample of a substrate, such as biomass, to an isothermal or non-isothermal heating program and recording the mass-loss rate due to temperature vs. time or temperature. Then, a differentiated thermogravimetric curve (DTG) and second derivative curve (DDTG) can be obtained from the experimental, which can provide an accurate profile of the thermal degradation of the biomass and its characteristics as feedstocks [55–57]. Based on the temperature ranges where the main degradations occur as well as other data such as the initial and residual mass and its variation, it is possible to establish the kinetics of biomass pyrolysis and determine the activation energy of the decomposition reactions of its main components using several previously-developed model-based methods [58–60]. Additionally, higher heating value (HHV) and ash content (AC) represent the most important parameters for discriminating solid biofuel quality. The presence of lignin increases HHV, but it can also decrease with high AC, while moisture content (MC) reduces lower heating value (LHV). All these parameters

influence energy generation performance and its economic aspects. Additionally, the presence of other elements such as nitrogen, sulfur, chlorine and some minor elements can be especially relevant for environmental aspects [61,62].

## 2. Materials and Methods

In this study, 15 samples of the most common European woody biomass species used in the energy pellet sector were selected, collected and classified into 3 categories: hardwood, softwood and chemically-treated wood. The samples shown in Table 1 were ground using a cutting mill (model SM 2000, Retsch), sieved below 0.25 mm and stored in plastic containers for subsequent analysis.

**Table 1.** The woody biomass samples used in this study.

| Category | Woody Biomasses | | | | |
|---|---|---|---|---|---|
| Hardwood | Common name<br>Scientific name<br>Abbreviation | Walnut<br>*Juglans regia*<br>Nut. | Sessile Oak<br>*Quercus petraea*<br>Oak | Chestnut<br>*Castanea sativa*<br>Chest. | Ash<br>*Fraxinus*<br>Ash | Beech<br>*Fagus sylvatica*<br>Beech |
| Softwood | Common name<br>Scientific name<br>Abbreviation | Larch<br>*Larix Decidua*<br>Larch | Pine<br>*Pinus*<br>Pine | Fir<br>*Abies*<br>Fir | Juniper<br>*Juniperus communis*<br>Jun. | Douglas Fir<br>*Pseudotsuga menziesii*<br>Dou. |
| Chemically-Treated Wood | Common name<br>Scientific name<br>Abbreviation | Medium density Fibreboard<br>-<br>MDF | Oriented Strand Board<br>-<br>OSB | Chipboard<br>-<br>Chip. | Laminated Pine<br>-<br>L. Pine | Laminated Ash<br>-<br>L. Ash |

Thermogravimetric analyses were performed using a TGA LINSEIS model STA PT-1600 in nitrogen (flow 100 mL/min) with alumina crucibles by heating 30 °C/min up to 110 °C, isothermal for 30 min and further heating of 5 °C/min up to 700 °C. The first drying steps were omitted from the results. Carbon, hydrogen, nitrogen and oxygen content was determined following the ISO 16948 (E) standard (Solid Biofuels-determination of total content of carbon, hydrogen and nitrogen) using an elemental analyzer (TruSpec CHN, Leco). In these analyses, about 70 mg of wood powder was burned at 950 °C (for about 5 min) under an oxygen atmosphere. At this temperature, all forms of nitrogen were oxidized to NOx. After humidity and ash elimination, the concentrations of $NO_2$ and NOx were determined by thermal conductivity. The higher heating value (HHV) was also determined in accordance with EN ISO 18125 (Solid biofuels-determination of calorific value). Following this method, we burned a mass of $1.0 \pm 0.2$ g of woody biomass in high-pressure oxygen in a bomb calorimeter (Isoperibolic calorimeter mod.C2000 basic, IKA). Then, the lower heating value was calculated considering moisture and hydrogen content.

Additionally, we determined the ash content by following the ISO 18122 (E) standard (Solid biofuels—determination of ash content). The porcelain crucible containing 1.0 g of woody biomass was burnt in a TGA Leco 701 at 550 °C for at least two hours. Instruments and methods are summarized in Table 2.

**Table 2.** Instruments and methods for the physical and chemical characterization of woody biomass.

| Analyses | Instrument | Method |
|---|---|---|
| Thermogravimetry | TGA LINSEIS (STA PT-1600) | - |
| Elemental Composition (CHNO) | Elemental analyser (TruSpec CHN, Leco) | ISO16948 |
| Calorific Value | Isoperibolic calorimeter mod.C200 basic, IKA | ISO18125 |
| Ash Content | TGA Leco 701 | ISO18122 |

Since our purpose is not to evaluate each woody species individually, but to compare the chemical-physical characteristics and the thermal behavior of the different types of biomasses, every sample was analyzed in a single test.

## 3. Results

### 3.1. Thermogravimetric Analysis Results

The thermograms (TG, DTG and DDTG) obtained show the thermal degradation (pyrolysis) profiles of the woody biomass samples studied. From the TGA, it is possible to obtain the conversion rate, which differentiates with respect to time, represents the DTG curve and gives the mass-loss rate. The second derivative (DDTG) curve provides the inflection points, representing the significant changes in the mass-loss rates occurring at various temperatures or residence times [63]. We divided the thermograms into three zones: Zone I starts from the ambient temperature to around 150 °C and corresponds to the mass loss due to evaporation of water and light volatiles; Zone II, ranging from 150 °C to about 500 °C, represents the main pyrolysis stage and is caused by the devolatilization of hemicellulose, cellulose and lignin; Zone III, above 500 °C, is the zone in which a small mass loss occurs largely due to the degradation of carbonaceous in the residues.

The TG results in Table 3 and Figure 1 highlight the most significant mass-loss trend in Zone II, in which molecular weight compounds split into smaller molecular weight compounds by cause of the steady supply of thermal energy. In this zone, regardless of the type of biomass, a rapid and greater mass loss of around 70% was observed, since the residual mass decreased from an average of 91% at 150 °C to 18% at 650 °C. Although, some differences were more obvious at 650 °C, depending on the category of wood: chemically-treated wood, softwood and hardwood showed a residual mass of 20%, 18% and 16%, respectively. Additionally, residual mass differences were identified within each category, especially in chemically-treated woods. For example, while, at 650 °C, the average residual mass of laminated pine and laminated ash was around 15 %, the average residual mass of MDF, OSB and chipboard was about 23%. On the other hand, Figure 2 shows that the conversion ratio increases rapidly from 250 °C to 350 °C, reaching values of 0.80–0.85.

**Table 3.** Mass % at key temperature points of the thermogravimetric analysis results.

| Temp. (°C) | Hardwood | | | | | Softwood | | | | | Chemically-Treated Wood | | | | |
|---|---|---|---|---|---|---|---|---|---|---|---|---|---|---|---|
| | Nut | Oak | Chest. | Ash | Beech | Larch | Pine | Fir | Jun. | Dou. | MDF | OSB | Chip. | L. Pine | L. Ash |
| 150 | 92 | 93 | 86 | 94 | 93 | 92 | 94 | 92 | 86 | 91 | 94 | 93 | 94 | 89 | 88 |
| 500 | 23 | 22 | 21 | 23 | 19 | 25 | 23 | 23 | 20 | 21 | 28 | 27 | 28 | 19 | 17 |
| 650 | 19 | 16 | 13 | 19 | 15 | 21 | 19 | 19 | 16 | 16 | 23 | 22 | 24 | 15 | 14 |

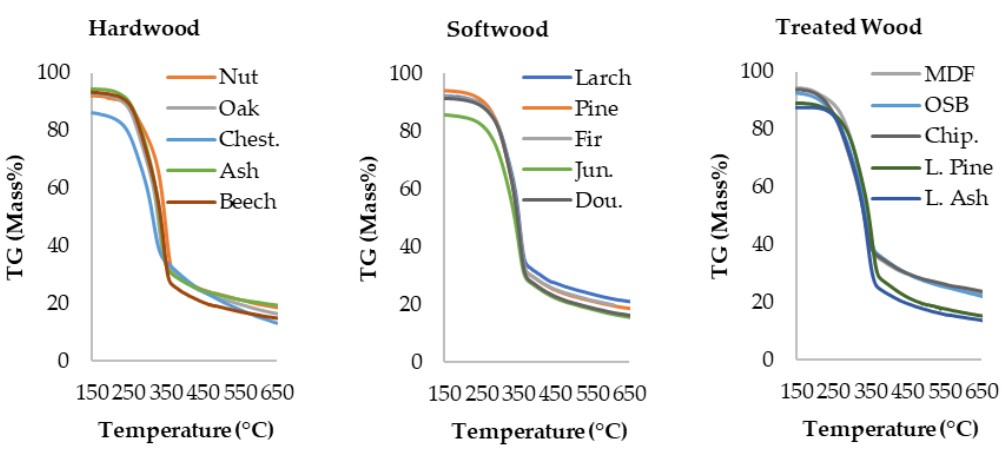

**Figure 1.** TG curves of thermal degradation of woody biomass samples.

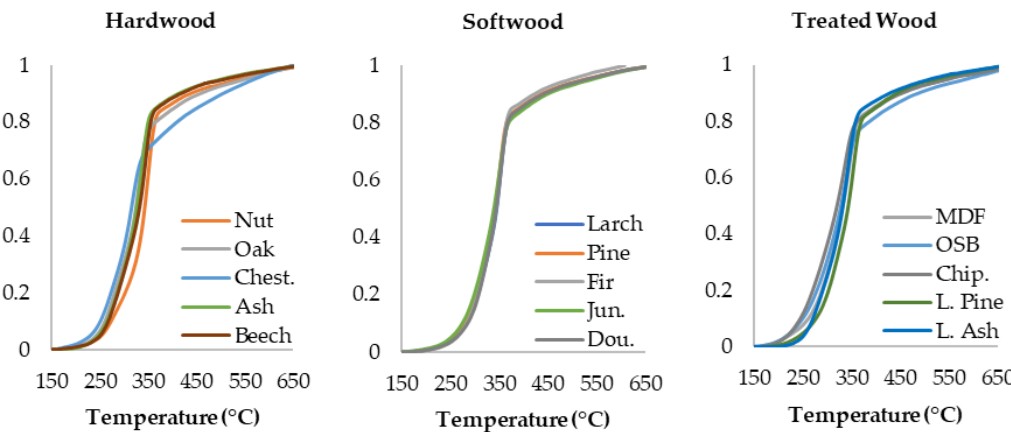

**Figure 2.** The conversion ratios of woody biomass samples.

Despite the interactions between the fractions of woody biomass samples, in the DTG curves (Figure 3) it was possible to identify a shoulder, a peak and a long tailing caused by the conversion of hemicellulose, cellulose and lignin, respectively. It should be noted that, in the presence of oxygen (combustion), it might be possible to see in the thermogravimetric graph a fourth peak around 450 °C due to the combustion of char [64,65]. The initial and final points of these three stages were identified by using the second derivative of the TG curve or differential DTG (DDTG) curves (Figures 4 and 5) [66]. Hemicellulose breaks down rapidly and the end point of the shoulder of the DTG curve, which varies from 280 °C to 320 °C, is the local minimum of the DDTG curve. The rapid conversion of hemicellulose is due to its composition, which consists primarily of five-carbon sugars such as xylose and arabinose in hardwoods and six-carbon sugars such as glucose, mannose and galactose in softwoods. Polysaccharides derived from these monomeric units have low thermal stability and are easily subject to hydrolysis and dehydration reactions [67,68]. For example, the decomposition of xylans, whose typical content is 10–35% in hardwoods and 10–15% in softwoods, yield mainly water, formic, hydroxy-1-propanone, hydroxy-1-butanone, methanol, 2-furfuraldeyde and acetic, and propionic acids that can be found in bio-oils [69]. Although the breakdown of cellulose starts below 280 °C with reactions such as dehydration, free radical formation and the construction of oxygenated moieties (such as carbonyls, carboxyl groups and peroxides), the pyrolysis of cellulose corresponds mainly to the second stage of DDTG, which shares the peak of the DTG curve. At this stage, the breakdown of cellulose results in a tar-rich pyrolysate encompassing levoglucosan, anhydrosugars, oligosaccharides, some glucose decay compounds due to depolymerization glycosidic bond, lower molecular weight gases and volatile products [70]. The DTG curve showed that the position of the peaks was quite similar for all the softwood samples, and the maximum mass-loss rate was around 350 °C, while the hardwood samples showed peaks in various positions and the maximum mass-loss rate ranged between 315 °C and 350 °C. The peaks of the chemically-treated wood samples showed less variation in their position, and the maximum mass-loss rate ranged between 340 °C and 355 °C. In addition, within each category, some differences in the maximum mass-loss rate were identified: chestnut showed a lower maximum mass-loss rate (around 0.75) compared to the other samples of hardwood studied (greater than 1); juniper also showed a lower maximum mass-loss rate (about 0.85) compared to the other samples of softwood studied (around 1.1); meanwhile, in chemically-treated wood, chipboard and OSB showed a lower maximum mass-loss rate compared to MDF, laminated pine and laminated ash. Additionally, the greater area of the shoulders and peaks of some wood samples would indicate a higher hemicellulose and cellulose content. The final temperature of the second stage on the DDTG curve, and, consequently, the initial point of the third stage, where lignin continues to break down at a slower rate, is the corresponding temperature at which $-d^2\,m/dT_c^2$ values no longer change or change very little (around 400 °C for most of the materials studied). The most appreciated depolymerization products from lignin fraction are high-

value aromatic monomers such as BTX (benzene, toluene and the three xylene isomers) and phenols, which can be used directly as basic fuels (e.g., bio marine fuel) or, such as holocellulose residue, can be catalytically upgraded to a wide variety of high-quality fuels and chemicals [71–73]. BTX is produced today in large amounts from fossil raw materials and is among the top 15 petrochemicals in terms of market, mainly because it is applied as building blocks for producing many secondary intermediates and final products [74,75]. Similar to depolymerization products from lignin fraction, holocellulose-rich pulp residue can be upgraded to a wide variety of high-quality fuels and chemicals [76,77].

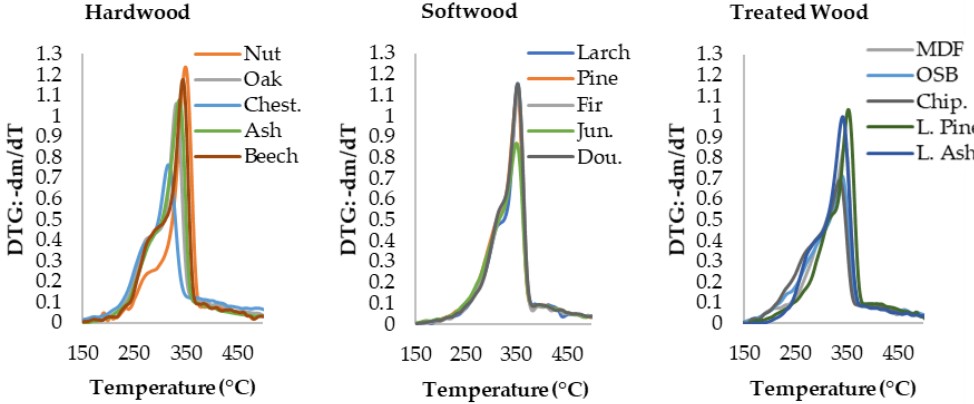

**Figure 3.** DTG curves of thermal degradation of woody biomass samples.

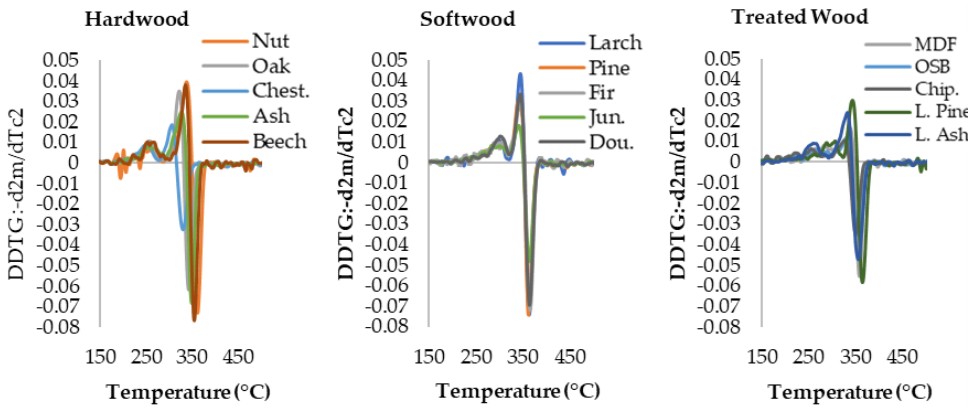

**Figure 4.** DDTG curves of thermal degradation of woody biomass samples.

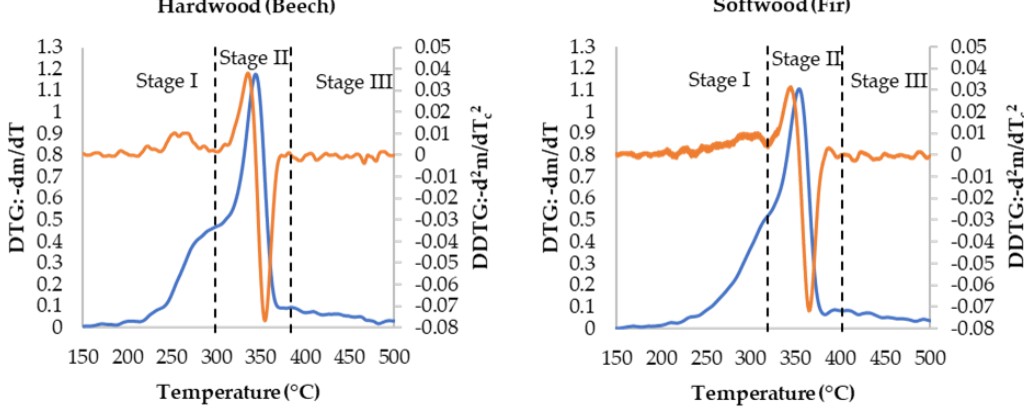

**Figure 5.** Examples of the Zone II division of the degradation profiles of hardwood and softwood samples using the second derivative of the TG curve method.

### 3.2. Physical and Chemical Characterization Results

Table 4 shows the elemental composition analyses of the woody biomass samples, which indicated that hardwood and softwood had a similar and higher carbon content than chemically-treated wood; particularly, MDF, OSB and chipboard showed the lowest carbon content. The average carbon content decreased in the order of softwood (50.74%) > hardwood (49.47%) > chemically-treated wood (47.46%). Additionally, from the results shown in Figure 6 and Table 5, the atomic O/C ratio of chemically-treated woods varied widely (from 0.59 to 0.75) compared to the O/C ratio of hardwoods (from 0.58 to 0.70) and softwoods (from 0.59 to 0.65). Likewise, we observed that the ash content of chemically-treated woods varied broadly (from 0.22% to 3.00%) compared to the O/C ratio of hardwoods (from 0.37% to 0.60%) and softwoods (from 0.37% to 0.60%). In general, the samples of hardwoods, softwoods and chemically-treated woods showed a similar energy content with HHV averages of 19.68 MJ/Kg, 19.42 MJ/kg and 19.59 MJ/kg, respectively. However, it is known that the carbon, oxygen and ash content could influence the calorific value of a fuel. A high H/C ratio, low O/C ratio and low ash content would provide a higher calorific value [78–80]. In Figure 6, this relationship was observed mainly in chemically-treated woods. For example, laminated pine and laminated ash, which had a lower O/C atomic ratio and lower ash content compared to MDF, OSB and chipboard, showed higher energy content. The energy content could also be related to the content of hemicellulose and cellulose, since the lower energy contents of MDF, OSB and chipboard agree with the lower areas of shoulders and peaks of the results of the thermogravimetric analyses. It should be noted that the ash is formed by the inorganic elements of the biomass and makes up the non-combustible part, which can generate the formation of slag and fouling at the end of the thermal degradation, as well as the corrosion of metallic surfaces of the burner systems such as boilers, furnaces, domestic stoves and gasifiers, consequently increasing the maintenance costs [81,82]. In addition, based on the atomic ratios (H/C, N/C and O/C), it is possible to elucidate the empirical formula for the woody biomass samples. For example, the empirical formula of MDF, OSB, chipboard, laminated pine and laminated ash could be described as $CH_{1.70}N_{0.06}O_{0.75}$, $CH_{1.73}N_{0.05}O_{0.71}$, $CH_{1.73}N_{0.09}O_{0.75}$, $CH_{1.41}N_{0.00}O_{0.59}$ and $CH_{1.48}N_{0.00}O_{0.67}$, respectively [83].

The samples of chemically-treated woods were also characterized by higher nitrogen content compared to hardwoods and softwoods, for which the nitrogen content was almost negligible. The average nitrogen content decreased in the order, chemically-treated wood (2.17%) > hardwood (0.17%) > softwood (0.13%). Notably, MDF (3.28%), OSB (2.47%) and chipboard (4.47%) showed the highest nitrogen content. The high nitrogen content in chemically-treated wood was mainly due to the presence of thermosetting adhesives. In the past, bio-based adhesives were used to glue wood. However, since the beginning of the 20th century, the market for thermosetting adhesives such as urea-formaldehyde, melamine-formaldehyde and isocyanate-based polymers has grown rapidly because they are generally considered more effective, less expensive and more stable in wet conditions [84,85]. Currently, urea-formaldehyde-based adhesives are almost exclusively used to produce wood-based materials such as OSB or MDF, mainly due to their low production cost, the versatility of use, high dry bond strength and colorless glue line. Although thermosetting adhesives are rich in nitrogen, they do not contain sulfur. Therefore, due to the very low sulfur content of woody biomass and the non-use of sulfur-based treatments in chemically-treated wood, the sulfur content could be less relevant in some cases for this type of material [86]. A lower sulfur content in solid biofuels could reduce corrosion problems in boilers and pipes, but it should be noted that feedstocks with a larger amount of nitrogen and sulphur lead to the production of NOx/SOx emissions [87].

Nitrogen is also released to a small extent as molecular nitrogen, nitrogen oxides and aromatic organic compounds [88]. These emissions negatively impact the acidification of the environment and climate change, increasing the global temperature disturbances and anomalies of the climatic phenomena and causing significant damage to the quality of ecosystems and biodiversity. Overall, it was identified that hardwoods, softwoods

and some chemically-treated woods, such as laminated pine and laminated ash, showed characteristics that comply with solid biofuel standard limits for high-quality pellet classes (ISO 17225-2:2021), i.e., ash content and heating value that comply with the limit of class A0.5 and nitrogen content that complies with the limit of class N0.2. Meanwhile chemically-treated woods such as MDF, OSB and chipboard showed an ash content and heating value that comply with the limit of class A1.

**Table 4.** Elemental composition and ash content of woody biomass samples.

| | Hardwood | | | | | | Softwood | | | | Chemically-Treated Wood | | | | |
|---|---|---|---|---|---|---|---|---|---|---|---|---|---|---|---|
| | **Nut** | **Oak** | **Chest.** | **Ash** | **Beech** | **Larch** | **Pine** | **Fir** | **Jun.** | **Dou.** | **MDF** | **OSB** | **Chip.** | **L. Pine** | **L. Ash** |
| C (%) | 52.10 | 49.10 | 49.20 | 48.24 | 48.40 | 50.07 | 50.55 | 49.86 | 52.49 | 50.70 | 45.20 | 46.10 | 44.50 | 52.36 | 49.16 |
| H (%) | 6.53 | 6.05 | 5.80 | 5.98 | 5.95 | 5.96 | 6.08 | 6.16 | 6.14 | 6.26 | 6.45 | 6.70 | 6.48 | 6.21 | 6.12 |
| N (%) | 0.33 | 0.11 | 0.20 | 0.21 | 0.17 | 0.23 | 0.12 | 0.12 | 0.15 | 0.11 | 3.28 | 2.47 | 2.87 | 0.10 | 0.13 |
| O (%) | 40.57 | 44.25 | 44.2 | 45.00 | 45.13 | 43.52 | 43.04 | 43.66 | 40.92 | 42.77 | 45.07 | 43.87 | 43.12 | 41.11 | 44.06 |
| Ash (%) | 0.50 | 0.50 | 0.60 | 0.57 | 0.35 | 0.22 | 0.20 | 0.20 | 0.30 | 0.16 | 1.00 | 0.86 | 3.03 | 0.22 | 0.53 |

**Table 5.** Atomic ratios and energy content of woody biomass samples.

| | Hardwood | | | | | | Softwood | | | | Chemically-Treated Wood | | | | |
|---|---|---|---|---|---|---|---|---|---|---|---|---|---|---|---|
| | **Nut** | **Oak** | **Chest.** | **Ash** | **Beech** | **Larch** | **Pine** | **Fir** | **Jun.** | **Dou.** | **MDF** | **OSB** | **Chip.** | **L. Pine** | **L. Ash** |
| H/C | 1.49 | 1.47 | 1.40 | 1.47 | 1.46 | 1.41 | 1.43 | 1.47 | 1.39 | 1.47 | 1.70 | 1.73 | 1.73 | 1.41 | 1.48 |
| N/C | 0.01 | 0.00 | 0.00 | 0.00 | 0.00 | 0.00 | 0.00 | 0.00 | 0.00 | 0.00 | 0.06 | 0.05 | 0.09 | 0.00 | 0.00 |
| O/C | 0.58 | 0.68 | 0.67 | 0.70 | 0.70 | 0.65 | 0.64 | 0.66 | 0.59 | 0.63 | 0.75 | 0.71 | 0.75 | 0.59 | 0.67 |
| HHV (MJ/Kg) | 20.02 | 19.62 | 19.20 | 19.74 | 19.82 | 19.40 | 19.27 | 19.42 | 19.56 | 19.47 | 18.58 | 19.25 | 19.13 | 21.02 | 19.98 |
| LHV (MJ/Kg) | 18.60 | 18.30 | 17.88 | 18.44 | 18.51 | 18.10 | 17.92 | 18.08 | 18.23 | 18.10 | 17.34 | 17.98 | 17.78 | 18.57 | 18.64 |

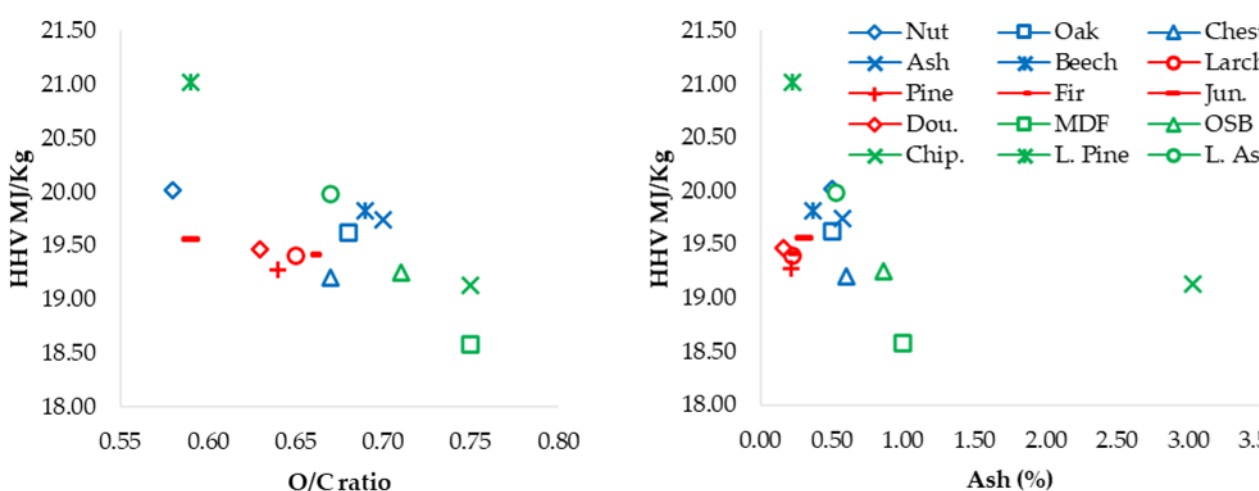

**Figure 6.** Relationship between HHV, O/C ratio and ash content of woody biomass samples.

## 4. Conclusions

This study evaluated the thermal behavior and chemical-physical characteristics of three woody biomass species (hardwood, softwood and chemically-treated wood) using thermogravimetric analysis and characterization analysis. The results showed that the different woody biomass species had an energy content ranging between 18.58 MJ/kg and 21.02 MJ/kg, and that the most significant trend of mass loss in the thermal degradation, around 70%, occurred between 150 °C and 500 °C, while the residual mass at 650 °C was between 13% and 24%. Additionally, it was evidenced that the softwood samples had a more stable thermal degradation compared to hardwoods and chemically-treated woods, mainly due to the different content of hemicellulose, cellulose and lignin. These characteristics make the woody

biomass species studied optimal energy sources that can be used directly or indirectly in low emission generation technologies that are currently available on the market such as combined heat and power (CHP) systems or technologies that are in the phase of development, such as fuel cells, in addition to being excellent resources to produce value-added derivatives, so their use can be extended to many other industrial sectors. However, to select the most suitable woody biomass species, not only should its quality as fuel be considered, but also other factors such as economic and environmental factors or the availability of the feedstock in a specific area, which will be decisive by promoting new investments in the market of clean energy generation and biorefineries.

**Author Contributions:** Conceptualization, M.J.A.R. and G.T.; methodology, M.J.A.R.; validation, M.J.A.R. and G.T.; investigation, M.J.A.R.; resources, G.T. and D.D.; data acquisition, V.M., S.D.S. and C.D.F.; writing—original draft preparation, M.J.A.R.; writing—review and editing, M.J.A.R., D.D. and G.T.; supervision, G.T.; project administration, G.T.; funding acquisition, G.T. and D.D. All authors have read and agreed to the published version of the manuscript.

**Funding:** This research received no external funding.

**Data Availability Statement:** Not applicable.

**Conflicts of Interest:** The authors declare no conflict of interest.

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
