# Peer review of "Preliminary Study on the Thermal Behavior and Chemical-Physical Characteristics of Woody Biomass as Solid Biofuels"

_processes, doi:10.3390/pr11010154_

Round 1

Reviewer 1 Report

The paper presents results of three standard laboratory analyses of solid fuels and briefly discusses the perspectives of the three raw materials undertaken, in thermal processes. It also includes some fundamental theory on fuel applications. Since the results may be useful for future use in applications of thermal processes, the paper is worth to be published.

The paper is generally at a fundamental level. The text is written in acceptable English without any syntax errors.

Two basic changes are needed:

1.       The abstract is more an introduction than an abstract. It must be rephrased including scope, materials, methods and results.

2.       An essential part of any paper is a brief bibliographic review that demonstrates the needs for further research in the subject under consideration. In this way the scope of the paper is justified. This is missing and must be added to the paper.

Then, the paper can be published.

Author Response

REVIEWER 1:

The paper presents results of three standard laboratory analyses of solid fuels and briefly discusses the perspectives of the three raw materials undertaken, in thermal processes. It also includes some fundamental theory on fuel applications. Since the results may be useful for future use in applications of thermal processes, the paper is worth to be published.

The paper is generally at a fundamental level. The text is written in acceptable English without any syntax errors.

Two basic changes are needed:

  1. The abstract is more an introduction than an abstract. It must be rephrased including scope, materials, methods and results.

The abstract was rephrased and information about scope, materials, methods and results was added.

  1. An essential part of any paper is a brief bibliographic review that demonstrates the needs for further research in the subject under consideration. In this way the scope of the paper is justified. This is missing and must be added to the paper.

The introduction was reformulated, highlighting the need to increase knowledge about the thermal behavior and physical-chemical characteristics of woody biomass to use it more efficiently. Not only in large-size generation plants, but also for example, in micro- and small-scale combined heat and power (CHP) systems for small- and medium-sized enterprises (SMEs) and residential applications, which could be a strategy to achieve a greater decentralization and rapid transition towards decarbonization of the electricity sector

Then, the paper can be published.

Reviewer 2 Report

The manuscript presents characterization of woody biomasses for micro- and small-scale CHP systems. While there is no particularly great novelty to the work as such, additional studies such as this none the less add to the body of literature available on a variety of biomasses, and as long as the work is sound and reported well, warrants publication.

Unfortunately, the manuscript in it's current form is far from acceptable and needs considerable work, in some ways even determining the soundness of the actual work is hard (thus "no answer" on that category at this stage). Some of the main points are listed in the following.

1. The article talks about woody biomasses for micro- and small-scale CHP, but the link between the characterization study performed, and specific use in small/micro-scale CHP appears forced. Why would characterization be useful only for <1 MWth scale CHP use? Either make the link clear, or perhaps better yet, just drop the micro/small CHP aspect and focus on woody biomass characterization. In the latter case, expand on the current and potential future uses and conversion paths for woody biomass, and the role the authors see for such materials in energy use and climate change mitigation.

2. Chapter 3.1: whether the small CHP aspect is kept or not, this chapter is not describing results of the study and difficult to justify in chapter 3, but extended background or a shallow attempt at a review.
Consider to what extent if any this material would be kept in the manuscript, perhaps appended to the introduction, and re-write with focus on woody biomass, or woody biomass for micro/small CHP, depending on which route the authors take regarding point 1.
There is an excessively long paragraph on page 5 - structure better to smaller paragraphs. Discussion on anaerobic digestion (not usual for woody biomass) should be considerably shortened/cut entirely, or explained why the authors think this is or will become important for woody biomass use, for small CHP or otherwise.
Fig.2 is grossly incomplete and simplistic as is - if kept in the article, include missing conversion paths (torrefaction, slow pyrolysis, HTC, ...) and products (e.g. char) and what are the main and secondary products from which conversions. If the angle of small/micro CHP is kept, a properly made version of this figure could be used to indicate which conversion processes and which energy carriers from those could be relevant for such use.

3. The experimental process
3.1 How was the oxygen content determined? By difference? If so, then was sulphur simply assumed zero?
If so - while sulphur content is of course likely to be very very small (although non-zero) for the untreated wood, it is less obvious for the treated wood. This should really be measured, or at the very least discussed in detail with references what magnitudes of values could be expected for each category of wood.
 3.2 I did not notice how many times the authors performed the experiments? In triplicate? As biomass samples are not homogeneous, hopefully not once? Number of experiments, and standard deviation should be clearly reported, not just the average results.

4. Reporting the experiments. Assuming that the experiments were properly carried out (not yet possible to determine), this is the weakest point of the manuscript. The article is not long, and by compacting the earlier parts along the lines suggested in previous points, the authors can take considerably more space for presenting the results in clear format.
The elemental compositions are poorly reported; attempting to do so with only a non-logarithmic line chart with magnitudes around 0.5, 5 and 50% for N, H and C&O is hopeless.
First of all, a table of the elemental composition and ash content is needed, and second, if additionally to the table or tables a figure or figures (perhaps splitting for soft/hard/treated woods is worth considering for clearer figures?), think how to present the data in a way that is useful, clear and easily readable. Consider bar charts for CHON & ash bars separately for each, maybe logarithmic axis, perhaps numerical values of each data point.
Make sure that with each table you know why it's there, what it tells, and that the way you made it is a good way to present clearly what you want to present.

5. Conclusions are shallow and superficial. Regarding both results, discussion and conclusions : think carefully what was done, why, what does it mean and what all conclusions can be drawn, and how do you best present it to the reader.

6. Finally, as a minor point, in parts of the manuscript the english language is sloppy, with mistakes, and the overly complex sentence structures make reading tedious at times.

Author Response

The manuscript presents characterization of woody biomasses for micro- and small-scale CHP systems. While there is no particularly great novelty to the work as such, additional studies such as this none the less add to the body of literature available on a variety of biomasses, and as long as the work is sound and reported well, warrants publication.

Unfortunately, the manuscript in it's current form is far from acceptable and needs considerable work, in some ways even determining the soundness of the actual work is hard (thus "no answer" on that category at this stage). Some of the main points are listed in the following.

  1. The article talks about woody biomasses for micro- and small-scale CHP, but the link between the characterization study performed, and specific use in small/micro-scale CHP appears forced. Why would characterization be useful only for <1 MWth scale CHP use? Either make the link clear, or perhaps better yet, just drop the micro/small CHP aspect and focus on woody biomass characterization. In the latter case, expand on the current and potential future uses and conversion paths for woody biomass, and the role the authors see for such materials in energy use and climate change mitigation.

Section 3.1 “micro- and small-scale CHP systems” was synthesized and moved the introduction. Additionally, information about the potential future uses of woody biomass was added (from line 265 to line 293).

  1. Chapter 3.1: whether the small CHP aspect is kept or not, this chapter is not describing results of the study and difficult to justify in chapter 3, but extended background or a shallow attempt at a review.

Consider to what extent if any this material would be kept in the manuscript, perhaps appended to the introduction, and re-write with focus on woody biomass, or woody biomass for micro/small CHP, depending on which route the authors take regarding point 1.

Section 3.1 “micro- and small-scale CHP systems” was synthesized and moved the introduction.

There is an excessively long paragraph on page 5 - structure better to smaller paragraphs. Discussion on anaerobic digestion (not usual for woody biomass) should be considerably shortened/cut entirely, or explained why the authors think this is or will become important for woody biomass use, for small CHP or otherwise.

The discussion about anaerobic digestion was eliminated. Initially, this information was considered because it was a way of explaining one of the biochemical processes used to transform solid biomass into a gaseous fuel (biogas) suitable for internal combustion engines.

Fig.2 is grossly incomplete and simplistic as is - if kept in the article, include missing conversion paths (torrefaction, slow pyrolysis, HTC, ...) and products (e.g. char) and what are the main and secondary products from which conversions. If the angle of small/micro CHP is kept, a properly made version of this figure could be used to indicate which conversion processes and which energy carriers from those could be relevant for such use.

The Fig. 2 was eliminated.

  1. The experimental process

3.1 How was the oxygen content determined? By difference? If so, then was sulphur simply assumed zero?

Yes, the oxygen content was determined by difference.

If so - while sulphur content is of course likely to be very very small (although non-zero) for the untreated wood, it is less obvious for the treated wood. This should really be measured, or at the very least discussed in detail with references what magnitudes of values could be expected for each category of wood.

The high nitrogen content in chemically treated wood was mainly due to the presence of thermosetting adhesives. However, although these compounds are rich in nitrogen, they do not contain sulfur. Therefore, considering the very low sulfur content in woody biomass and the non-use of sulfur-based treatments in chemically treated wood, the sulfur content was not estimated. Literature supporting this approach was added (from line 298 to line 310).

3.2 I did not notice how many times the authors performed the experiments? In triplicate? As biomass samples are not homogeneous, hopefully not once? Number of experiments, and standard deviation should be clearly reported, not just the average results.

The analysis about the virgin samples were performed once because we can assume the material is homogeneus. Infact, as reported in Material and Methods, the samples were ground by means of a cutting mill (model SM 2000, Retsch), sieved below 0.25 mm. This procedure creates a homogeneus material and according to our experience the showed results are reliable. The chemical treated woods, because of the presence of thermosetting adhesives, are lower homogeneus materials. In this case we preferred to report only the most reliable results. The authors are aware that the biomass samples are not homogeneous, the analyzes were carried out two or three times, however, not all the results were reliable, so a review was carried out and only the reliable results were selected, one per woody biomass sample.

  1. Reporting the experiments. Assuming that the experiments were properly carried out (not yet possible to determine), this is the weakest point of the manuscript. The article is not long, and by compacting the earlier parts along the lines suggested in previous points, the authors can take considerably more space for presenting the results in clear format.

Although the study shows results from only one analysis per sample of woody biomass, this does not mean that the analyzes have been carried out inappropriately. In fact, in the manuscript the methodology used for the thermogravimetry analyzes and for the physical and chemical characterization is declared (ISO Standards). Some parts of the manuscript were rephrased and improved, as well as figures and tables, and additional literature was added.

The elemental compositions are poorly reported; attempting to do so with only a non-logarithmic line chart with magnitudes around 0.5, 5 and 50% for N, H and C&O is hopeless.

First of all, a table of the elemental composition and ash content is needed, and second, if additionally to the table or tables a figure or figures (perhaps splitting for soft/hard/treated woods is worth considering for clearer figures?), think how to present the data in a way that is useful, clear and easily readable. Consider bar charts for CHON & ash bars separately for each, maybe logarithmic axis, perhaps numerical values of each data point.

Make sure that with each table you know why it's there, what it tells, and that the way you made it is a good way to present clearly what you want to present.

Figure 8 was replaced by Table 4 which reports the results of the elemental composition and ash content of woody biomass samples. We think that it is not necessary to add other figures since they would repeat the same results of the table.

  1. Conclusions are shallow and superficial. Regarding both results, discussion and conclusions : think carefully what was done, why, what does it mean and what all conclusions can be drawn, and how do you best present it to the reader.

The authors believe that the conclusions reflect the results obtained. However, some parts of the manuscript were reformulated and improved to better present the results to the reader.

  1. Finally, as a minor point, in parts of the manuscript the english language is sloppy, with mistakes, and the overly complex sentence structures make reading tedious at times.

In some parts of the manuscript the English language was corrected to avoid a possibly tedious reading.

Reviewer 3 Report

Dear authors,

After reviewing your manuscript have to express some concerns about it.

1). The first part of the results, section 3.1 Micro- and small-scale CHP systems, does not contain new achievements or new information provided by you. Thus, this section seems more like a literature review and should be in the introduction section.

2). There is a lack of clarity in the comparative analysis of the higher heating value among wood samples, for example, the direct relation between the O/C ratio and HHV or ash% and HHV.

3). The title is “Characterization of woody biomasses and their use as fuel in 2 micro- and small-scale CHP systems”, and the investigation presents only the first part, characterization of woody biomasses. Thus, the use as fuel is not analyzed experimentally nor theoretically, which is the critical application of biomass.

4). Using these biomasses in CHP systems is crucial to analyze their emissions. Otherwise, the characterization will be incomplete.

5). If you do not analyze the use of these biomasses as fuel in CHP systems experimentally or theoretically, it should not be part of the title.

6). You must review all the figures using commas for decimals, which should be a dot. Also, in Figure 9, the O/C ratio axis numbers are wrong. As a general recommendation, all the figures should be reviewed in other software to clarify, not in Excel.

Author Response

Dear authors,

After reviewing your manuscript have to express some concerns about it.

1). The first part of the results, section 3.1 Micro- and small-scale CHP systems, does not contain new achievements or new information provided by you. Thus, this section seems more like a literature review and should be in the introduction section.

Section 3.1 was synthesized and moved to the introduction

2). There is a lack of clarity in the comparative analysis of the higher heating value among wood samples, for example, the direct relation between the O/C ratio and HHV or ash% and HHV.

The section corresponding to the results of the analysis of elemental composition and energy content was reformulated to make it more understandable (from line 239 to line 253). Additionally Figure 7 "Relationship between HHV, O/C ratio and ash content of woody biomass samples" was improved.

3). The title is “Characterization of woody biomasses and their use as fuel in 2 micro- and small-scale CHP systems”, and the investigation presents only the first part, characterization of woody biomasses. Thus, the use as fuel is not analyzed experimentally nor theoretically, which is the critical application of biomass.

The title of the manuscript was changed to “Preliminary study about the thermal behavior and chemical-physical characteristics of woody biomass as solid biofuels”

4). Using these biomasses in CHP systems is crucial to analyze their emissions. Otherwise, the characterization will be incomplete.

We agree with the comment, possibly the title of the manuscript was not the most appropriate to reflect the work done.

5). If you do not analyze the use of these biomasses as fuel in CHP systems experimentally or theoretically, it should not be part of the title.

The title of the manuscript was changed to “Preliminary study about the thermal behavior and chemical-physical characteristics of woody biomass as solid biofuels”

6). You must review all the figures using commas for decimals, which should be a dot. Also, in Figure 9, the O/C ratio axis numbers are wrong. As a general recommendation, all the figures should be reviewed in other software to clarify, not in Excel.

All the figures were revised and improved, and a point was used instead of a comma to separate the decimals.

Reviewer 4 Report

1. Some quantification of results will be useful in the abstract

2. Is it possible to include cavitation based techniques also in the technologies (even if not as main technologies), what is your comment on cavitation based technology in the horizon

3. The figures are quite useful, but could be plotted in a better manner to bring out the key message usefully (for example Figure 3 could be made self sufficient by including the colour codes inside the figure), right now the figures do not look too impressive

In general the article has useful results, some improvement in presentation can improve the quality of presentation

Author Response

  1. Some quantification of results will be useful in the abstract

The abstract was rephrased and information about scope, materials, methods and results was added.

  1. Is it possible to include cavitation based techniques also in the technologies (even if not as main technologies), what is your comment on cavitation based technology in the horizon

The authors' knowledge about the use of cavitation techniques in power generation systems is limited, so inclusion of these techniques in the manuscript was avoided.

  1. The figures are quite useful, but could be plotted in a better manner to bring out the key message usefully (for example Figure 3 could be made self-sufficient by including the colour codes inside the figure), right now the figures do not look too impressive

All the figures were revised and improved.

In general, the article has useful results, some improvement in the presentation can improve the quality of presentation

Round 2

Reviewer 2 Report

The authors have modified the paper and in some ways improved it; unfortunately, the description of the experimental part of the work reveals that the study is in it’s current state too far from acceptable to be considered for publication. For this reason, I see no option but to recommend rejection at this stage, and re-submission once the shortcomings of the experiments have been resolved.

The major problem is the way the experiments were carried out and reported. The authors state that single experiments for virgin samples is reliable. This is a bold claim to make to put it mildly, particularly when some of the results are not what one would expect (e.g., only 0.02 % nitrogen in pine, an order of magnitude less than the 0.1 – 1.0 % range typical for wood biomasses).

I recommend carrying out the experiments in triplicate, and report the average and standard deviation, or at the absolute bare minimum two experiments, followed by third if there is large difference, throwing out whichever is the outlier. Make sure to include ALL results in supplementary material, including possible outliers.

The comment regarding the chemically treated woods reads even worse, the authors state ”the analyzes were carried out two or three times, however, not all the results were reliable, so a review was carried out and only the reliable results were selected, one per woody biomass sample”. This is a completely unscientific approach, and has a high chance of reporting what the authors think the result should probably be, rather than what is. The reader is left with no way to evaluate if the judgements were sound. It appears creating a homogeneous mix of the treated woods is difficult. The way around this is to carry out multiple experiments, and report the average, standard deviation, and confidence interval. Even triplicate may not be enough. Again, all results must be included as a supplementary material.

In addition to the main problem described above, also the following issues that should also be fixed when resubmitting:

1)      The language has been slightly improved at places, but the manuscript would benefit from a check by professional or native speaker.

2) The broad discussion of micro- and small-scale CHP in the introduction is still at odds with the actual research carried out, which was characterization of woody biomasses. The authors are clearly interested in small-scale bio-CHP production, which is fine, but the purpose of the introduction is to put the research in context. Deciding what to emphasize in the introduction should be based on which uses of woody biomasses are most important and widespread, and for which the characterization study is most useful and relevant.

3)     
Do think about the conclusions – as it is, this chapter is mostly a summary of the results, and the main conclusion is “considering the thermal behavior and chemical-physical characteristics of the woody biomass species studied, presumably it would be more suitable to use softwoods as fuel rather than hardwoods and chemically treated woods”. Firstly there are many ways to use woody biomass as a fuel, and secondly, availability and price often drive the fuel choice more than optimal characteristics. What problems result from the characteristics of the “less suitable” wood types, in what kinds of fuel use and pretreatments the issues would manifest themselves more, where less, how does one work around them?

Author Response

Reviewer 2

R2: The authors have modified the paper and in some ways improved it; unfortunately, the description of the experimental part of the work reveals that the study is in its current state too far from acceptable to be considered for publication. For this reason, I see no option but to recommend rejection at this stage, and re-submission once the shortcomings of the experiments have been resolved.

The major problem is the way the experiments were carried out and reported. The authors state that single experiments for virgin sample are is reliable. This is a bold claim to make to put it mildly, particularly when some of the results are not what one would expect (e.g., only 0.02 % nitrogen in pine, an order of magnitude less than the 0.1 – 1.0 % range typical for wood biomasses).

Authors: Considering the possible mistake identified about the nitrogen content, we decided to repeat the elemental and ash content analysis, in the manuscript the initial values were replaced by the new results, also correcting the H/C and O/C ratios.

R2: I recommend carrying out the experiments in triplicate, and reporting the average and standard deviation, or at the absolute bare minimum two experiments, followed by third if there is large difference, throwing out whichever is the outlier. Make sure to include ALL results in supplementary material, including possible outliers.

The comment regarding the chemically treated woods reads even worse, the authors state ”the analyzes were carried out two or three times, however, not all the results were reliable, so a review was carried out and only the reliable results were selected, one per woody biomass sample”. This is a completely unscientific approach, and has a high chance of reporting what the authors think the result should probably be, rather than what is. The reader is left with no way to evaluate if the judgements were sound. It appears creating a homogeneous mix of the treated woods is difficult. The way around this is to carry out multiple experiments, and report the average, standard deviation, and confidence interval. Even triplicate may not be enough. Again, all results must be included as a supplementary material.

In addition to the main problem described above, also the following issues that should also be fixed when resubmitting:

1)      The language has been slightly improved at places, but the manuscript would benefit from a check by professional or native speaker.

2) The broad discussion of micro- and small-scale CHP in the introduction is still at odds with the actual research carried out, which was characterization of woody biomasses. The authors are clearly interested in small-scale bio-CHP production, which is fine, but the purpose of the introduction is to put the research in context. Deciding what to emphasize in the introduction should be based on which uses of woody biomasses are most important and widespread, and for which the characterization study is most useful and relevant.

Authors: The analyzes were not carried out in triplicate because our aim was not to evaluate each biomass species individually, but to compare the chemical-physical characteristics and the thermal behavior of the different types of biomass. Due to economic and time limitations, we find it difficult to enlarge the working plan. We submitted the article to the review of an English native-speaker researcher who made several changes to the article, improving its quality

Micro- and small-scale CHP systems are the most widely used technologies globally in electricity generation. For this reason, the authors do not believe that dedicating part of the introduction to highlighting the different types of existing micro- and small-scale CHP systems and the potential routes for the sustainable use of biomass with these technologies is out of context.
On the other hand, the characterization of biomass is equally important for its use in any of the technologies described, and not only in the energy sector, but also in the industrial sector to obtain value-added products. Additional information about possible future uses of woody biomass was added from line 265 to line 293 of the manuscript. Readers could use the results obtained in this study to identify the most suitable type of biomass depending on the specific requirements.

R2: Do think about the conclusions – as it is, this chapter is mostly a summary of the results, and the main conclusion is “considering the thermal behavior and chemical-physical characteristics of the woody biomass species studied, presumably it would be more suitable to use softwoods as fuel rather than hardwoods and chemically treated woods”. Firstly there are many ways to use woody biomass as a fuel, and secondly, availability and price often drive the fuel choice more than optimal characteristics. What problems result from the characteristics of the “less suitable” wood types, in what kinds of fuel use and pretreatments the issues would manifest themselves more, where less, how does one work around them?

Authors: The aim of this article is the chemical characterization of woody biomasses and make a comparison of the different results. The authors are aware of the prices and other factors that might drive the fuel choice. But these considerations are beyond the scope of this article.

Reviewer 3 Report

The article has been significantly improved, and the information is more precise and better organized. Under these conditions, the work can be published.

Author Response

Thank you for your considerations.

Round 3

Reviewer 2 Report

The authors have made minor improvements to the manuscript, especially language has been improved (although there are still some typographical errors) but the major problems remain, and attempts to answer them unfortunately appear poor and downright evasive.

1. The main problem and the reason why at this stage the manuscript must be rejected, is that the results simply are not reliable enough. One test per sample is NOT sufficient. As stated before, ideally the experiments should be performed in triplicate but at a bare minimum, 2 tests, followed by 3rd if there is significant difference and whichever is the outlier thrown out. With the more heterogeneous synthetic materials, even this is probably not sufficient.

The  obviously incorrect results are now corrected, but the presence of such in the first place underlines that a single test is just not enough, and results are not reliable. If there are no resources to perform the required tests as the authors stated, my suggestion is to find a conference or some other channel than a scientific journal where to publish the results as they are.

2. The introduction still needs to be balanced. The authors state in the response that "Micro- and small-scale CHP systems are the most widely used technologies globally in electricity generation", which is a rather bemusing statement to make. For the reader, a brief introduction on global energy use of woody biomass, backed up by numbers from reputable references, would place the work in proper context; picking up one niche technology and expanding on that at excessive detail is unhelpful.

3. The conclusions are still short and superficial, and earlier comment was not addressed properly, so to repeat: as it is, this chapter is mostly a summary of the results, and the main conclusion is “considering the thermal behavior and chemical-physical characteristics of the woody biomass species studied, presumably it would be more suitable to use softwoods as fuel rather than hardwoods and chemically treated woods”. Firstly there are many ways to use woody biomass as a fuel, and secondly, availability and price often drive the fuel choice more than optimal characteristics. What problems result from the characteristics of the “less suitable” wood types, in what kinds of fuel use and pretreatments the issues would manifest themselves more, where less, how does one work around them?

This comment does NOT mean that the authors should expand on issues of availability or price - it means that issues other than optimal characteristics often drive the choice leading to use of what is available, not what is preferable, and the authors could use the questions as a help to expand discussion beyond simply stating that softwood is preferable.

Author Response

Reviewer 2

The authors have made minor improvements to the manuscript, especially language has been improved (although there are still some typographical errors) but the major problems remain, and attempts to answer them unfortunately appear poor and downright evasive.

1. The main problem and the reason why at this stage the manuscript must be rejected, is that the results simply are not reliable enough. One test per sample is NOT sufficient. As stated before, ideally the experiments should be performed in triplicate but at a bare minimum, 2 tests, followed by 3rd if there is significant difference and whichever is the outlier thrown out. With the more heterogeneous synthetic materials, even this is probably not sufficient.

The  obviously incorrect results are now corrected, but the presence of such in the first place underlines that a single test is just not enough, and results are not reliable. If there are no resources to perform the required tests as the authors stated, my suggestion is to find a conference or some other channel than a scientific journal where to publish the results as they are.

Answer: as already written in previous reviews, the authors consider the methodology sufficient for the purposes of the work. This concept has been discussed. Regarding the suggestion to publish in a conference the authors have a different point of view.

  1. The introduction still needs to be balanced. The authors state in the response that "Micro- and small-scale CHP systems are the most widely used technologies globally in electricity generation", which is a rather bemusing statement to make. For the reader, a brief introduction on global energy use of woody biomass, backed up by numbers from reputable references, would place the work in proper context; picking up one niche technology and expanding on that at excessive detail is unhelpful.

Answer: the introduction was rephrased, synthesizing the information about CHP systems. All references used in the manuscript are reputable.

3. The conclusions are still short and superficial, and earlier comment was not addressed properly, so to repeat: as it is, this chapter is mostly a summary of the results, and the main conclusion is “considering the thermal behavior and chemical-physical characteristics of the woody biomass species studied, presumably it would be more suitable to use softwoods as fuel rather than hardwoods and chemically treated woods”. Firstly there are many ways to use woody biomass as a fuel, and secondly, availability and price often drive the fuel choice more than optimal characteristics. What problems result from the characteristics of the “less suitable” wood types, in what kinds of fuel use and pretreatments the issues would manifest themselves more, where less, how does one work around them?

This comment does NOT mean that the authors should expand on issues of availability or price - it means that issues other than optimal characteristics often drive the choice leading to use of what is available, not what is preferable, and the authors could use the questions as a help to expand discussion beyond simply stating that softwood is preferable.

The conclusions were rephrased.